# Assessment of Factors Related to Diminished Appetite in Hemodialysis Patients with a New Adapted and Validated Questionnaire

**DOI:** 10.3390/nu13041371

**Published:** 2021-04-19

**Authors:** Elihud Salazar-Robles, Abel Lerma, Martín Calderón-Juárez, Armando Ibarra, Héctor Pérez-Grovas, Luis A. Bermúdez-Aceves, Lilian E. Bosques-Brugada, Claudia Lerma

**Affiliations:** 1Centro Universitario de la Costa, Universidad de Guadalajara, Puerto Vallarta 66376, Mexico; elihud.salazar@academicos.udg.mx; 2Instituto de Ciencias de la Salud, Universidad Autónoma del Estado de Hidalgo, Pachuca 42160, Mexico; abel_lerma@uaeh.edu.mx (A.L.); lilian_bosques@uaeh.edu.mx (L.E.B.-B.); 3Plan de Estudios Combinados en Medicina (PECEM), Facultad de Medicina, Universidad Nacional Autónoma de México, Mexico City 04510, Mexico; martin.cal.j@comunidad.unam.mx; 4Diálisis La Loma, Puerto Vallarta 48310, Mexico; renalvallarta@hotmail.com; 5Instituto Nacional de Cardiología Ignacio Chávez, Mexico City 14080, Mexico; hpgrovas@gmail.com; 6Centro Estatal de Hemodiálisis, Instituto Estatal de Cancerología, Colima 28000, Mexico; lbermudeza@hotmail.com

**Keywords:** hemodialysis, appetite, depression, anxiety, distorted thoughts

## Abstract

Appetite loss is a common phenomenon in end-stage renal disease (ESRD) patients undergoing maintenance hemodialysis (HD). We aimed to (i) adapt and validate a Spanish language version of the Council on Nutrition Appetite Questionnaire (CNAQ) and (ii) to identify psychological and biological factors associated with diminished appetite. We recruited 242 patients undergoing HD from four hemodialysis centers to validate the Spanish-translated version of the CNAQ. In another set of 182 patients from three HD centers, the Appetite and Diet Assessment Tool (ADAT) was used as the gold standard to identify a cut-off value for diminished appetite in our adapted questionnaire. The Beck Depression Inventory (BDI), Beck Anxiety Inventory (BAI), Distorted Thoughts Scale (DTS), Dialysis Malnutrition Score (DMS), anthropometric, values and laboratory values were also measured. Seven items were preserved in the adapted appetite questionnaire, with two factors associated with flavor and gastric fullness (Cronbach’s alpha = 0.758). Diminished appetite was identified with a cut-off value ≤25 points (sensitivity 73%, specificity 77%). Patients with diminished appetite had a higher proportion of females and DMS punctuation, lower plasmatic level of creatinine, blood urea nitrogen, and phosphorus. Appetite score correlated with BDI score, BAI score and DTS. Conclusions: This simple but robust appetite score adequately discriminates against patients with diminished appetite. Screening and treatment of psychological conditions may be useful to increase appetite and the nutritional status of these patients.

## 1. Introduction

Appetite loss is a common phenomenon in end-stage renal disease (ESRD) patients undergoing maintenance hemodialysis (HD) [1] and is a remarkable cause of protein–energy wasting syndrome [2]. Appetite loss is associated with symptoms of depression and anxiety [3], high concentrations of pro-inflammatory cytokines [4,5], hospitalization [6], diminished quality of life [7], and increased mortality [8]. Despite the high prevalence of nutritional disorders, there is not a standard method to assess appetite loss. The Appetite and Diet Assessment Tool (ADAT) is a nutrition-specific instrument [9] that has been used to measure anorexia prevalence in patients on hemodialysis [10]. A poor score is associated with adverse clinical outcomes such as hospitalization and risk of death [6].

Appetite is the state of motivation to eat [11], and it is subjected to several psychological and biological factors [12]. Previous studies have highlighted outage, gastrointestinal disturbances (e.g., gastric distention, constipation, impaired gastric emptying) [13,14], and even taste disturbances [15] as elements related to appetite loss in ESRD patients. The importance of the distinction of affective from somatic depressive symptoms in patients treated with hemodialysis has recently been reported due to the fact that affective symptoms predict 3-year mortality [16]. It is not clear whether appetite loss (a somatic symptom) is overlapped by uremia, inadequate hemodialysis, or psychiatric disturbances related to hemodialysis such as depression and anxiety [16]. 

On the other hand, appetite loss is a potential cause of the abovementioned complications and is intimately and bilaterally related to symptoms of depression and anxiety [3,17]. Furthermore, cognitive distortions are often neglected as a potential factor related to anorexia [18,19]. Addressing the factors related to low appetite and establishing its relationship with psychological symptoms may be relevant to improve the effectiveness of nutritional interventions. Particularly in Spanish-speaking countries, such as Mexico, where the incidence of ESRD is one of the most elevated in the world, the rate could be as high as 594 per million population, with 135 per thousand receiving dialysis [20].

The purpose of this work was to adapt and validate a Spanish language modified version of the Council on Nutrition Appetite Questionnaire [21], and additionally to investigate the psychological and biological factors associated with appetite loss. 

## 2. Materials and Methods

### 2.1. Patients and Study Protocol

The study included adult patients from three HD units located in two regions of Mexico: 242 patients who were enrolled for the validation of the appetite questionnaire (aged between 18 and 80 years old, 39% were women) and 182 patients (aged between 18 and 80 years old, 40% were women) who were enrolled for the analysis of factors related to low appetite. All patients had at least three months treatment with HD (3 times per week), with no hospitalization the month prior to the study. Clinical and biochemical variables from each patient were obtained from the clinical chart. Appetite, the Dialysis Malnutrition Score, the depression symptoms score, anxiety score, and the Distorted Thoughts Scale results were obtained from previously validated questionnaires. The protocol was approved by the Bioethics and Research Committees of the Instituto Nacional de Cardiología Ignacio Chávez (protocol number 17-1015). Each participant was informed of the nature of the study, agreed to participate, and signed a consent form.

### 2.2. Nutritional Status Evaluation

Body weight was taken on a midweek dialysis day after termination of treatment. Height was obtained from the patient’s chart to calculate the body mass index (BMI) value. The mid-arm circumference was measured with a plastic tape on the non-vascular access arm. Triceps skinfold thickness was measured with a conventional skinfold caliper using standard technique. The average of three measurements was obtained. The mid-arm muscle circumference was calculated using the next formula: mid-arm muscle circumference = mid-arm circumference—3.142 × triceps skinfold. Patients with protein energy waste syndrome were identified when they showed at least three of the following criteria: albumin <3.3 g/dL, body mass index <3.8 g, mid-arm muscle circumference <25.3 (male patients) or <23.2 (female patients), and normalized protein nitrogen appearance <0.8 g/kg/day [22,23]. Laboratory values (albumin, creatinine, hemoglobin, phosphorus, calcium, and blood urea nitrogen) were measured immediately before the dialysis in the midweek session. For assessment of the nutritional status, we used the Dialysis Malnutrition Score (DMS), which consists of 7 variables based on (a) a patient’s medical history: weight change, dietary intake, gastrointestinal symptoms, functional capacity, and comorbidity; and (b) a physical exam: loss of fat stores and signs of muscle wasting. Each DMS component has four levels of severity, from 1 (normal) to 5 (very severe). The sum of 7 components results in an overall score ranging from 7 (normal) to 30 (very severe). The DMS was applied by the same dietitian to all participants [24].

### 2.3. Appetite Assessment

The questionnaire used to assess appetite level was based on the Council on Nutrition Appetite Questionnaire, which includes 8 Likert-type items with a five-point scale for each [21]. The total appetite score ranges from 8 (worst appetite) to 40 (best appetite). The score was translated from English to Spanish by three independent experts to obtain a Spanish version.

We extracted a consecutive sample of 242 patients with ESRD from four hemodialysis units in four states in Mexico. For validation of the appetite questionnaire, we followed the method previously used to assess the psychometric properties of a questionnaire applied to patients with ESRD [25]. A randomized sample of 50% of this population was obtained in order to do an exploratory analysis followed by a confirmatory analysis as described below (Section 2.5). With the remaining 50% of participants, another confirmatory analysis was carried out to compare both models.

A receiver operator characteristic (ROC) curve analysis was performed on the total appetite score to identify a cut-off value for reduced appetite. We consider the Appetite and Diet Assessment Tool (ADAT) question as the gold standard [6,26]; that is, “During the past week, how would you rate your appetite?”, which has a 5-point Likert scale: (1) very good, (2) good, (3) fair, (4) poor, and (5) very poor. We considered an answer of “fair”, “poor”, or “very poor appetite” as diminished appetite in the new appetite scale [4].

### 2.4. Psychological Questionnaires

The Beck Depression Inventory (BDI) and Beck Anxiety Inventory (BAI) are scales used to measure depressive and anxiety symptoms in these ESRD patients [27,28,29]. They are both 21-item scales that address cognitive and somatic depression or anxiety symptoms, respectively, and are both widely used in general and clinical populations. The estimated reliability (Cronbach’s alpha) of the Mexican version of the BAI was 0.83 for a general population [30]. The reliability of the BDI was 0.87 for a general population [31]. The distorted thoughts scale (DTS) [18,19] was developed to assess irrational negative thinking in HD patients through 30 items comprising four subscales (perfectionism, catastrophism, negative self-labeling, and dichotomous thinking). The validation of this scale in a sample of 255 ESRD patients showed a Cronbach’s alpha of 0.93, and all subscales had positive correlations with both depression and anxiety symptoms [18].

### 2.5. Statistical Analysis

Since most variables did not have a normal distribution (Kolmogorov–Smirnov test), they were described by the median (percentiles 25–75%). Data were compared between patients with low appetite versus normal appetite with Mann–Whitney U tests or chi-squared tests. The variables independently associated with malnutrition were identified by multiple logistic regression analysis. A *p*-value < 0.05 was defined as statistically significant.

The validation of the appetite questionnaire was performed with an exploratory analysis via the following steps: (a) we compared the discriminative power of each item comparing the highest and lowest punctuations through a *t*-test; (b) the directionality of the items was assessed using crossed tables; (c) internal reliability was evaluated using Cronbach’s alpha statistical analysis; (d) orthogonal analysis was used to identify the items that belonged to each dimension of the scale; (e) the usefulness of the component structure was tested using the Kaiser–Meyer–Olkin (KMO) test (sample adequacy index) and Barlett’s sphericity test; (f) items that were located in two or more component dimensions of the scale were excluded [32].

Subsequently, we executed the confirmatory factorial analysis (CFA) by means of the maximum likelihood method, pondering a multifactorial model of two domains with the covariance of measurement errors. From the previous exploratory analysis, structural equations were estimated to calculate the following indices: Chi-squared test (X2), Chi-squared model index (CMIN), X2/degrees of freedom (CMIN/*df*), both for parsimony analysis of the model measures; adjusted goodness of fit index (GFI) and its complementary indices, the Tucker–Lewis (TLI) and the comparative goodness of fit index (CFI) indices; root mean square error of approximation (RMSEA) [33,34].

## 3. Results

### 3.1. Appetite Score Validation

#### 3.1.1. Exploratory Analysis

After strictly ensuring that there were no empty cells and checking the directionality in the crossed tables, as well as the discrimination capacity of all items (t-test with a *p*-value less than 0.001), item 8 (“Most of the time I feel...”) was eliminated from the questionnaire and the following analyses because it was located simultaneously within two factors, leaving seven items with factor loads of at least 0.40 that were distributed in two factors (factor 1 = 0.739, 3 items; factor 2 = 0.656, 4 items), Table 1. The total internal consistency of the questionnaire (Cronbach’s alpha) was α = 0.758, which explains 57.6% of the total variance; the average of the total scores = 24.6 ± 4, variance = 16.1, with an F value = 376.6, 116 *df*, *p* ≤ 0.000 in Hotelling’s t-square test. Bartlett’s sphericity test yielded an estimator of 181.9, 21 degrees of freedom (*df*), *p* < 0.001, confirming non-identity in the correlation matrix. On the other hand, the KMO indicator was close to the value of 1 (0.783), confirming the sample adequacy. The final version of the validated questionnaire is shown in Appendix A.

#### 3.1.2. Confirmatory Analysis

The following steps were applied under the maximum likelihood method to the 2 factors obtained in the previous exploratory analysis to evaluate their adjustment through a confirmatory factor analysis (CFA): (1) identification of the model through its components (endogenous and exogenous variables, as well as the unobserved variables to be identified by the model), whereby the total of the variables was less than the non-redundant elements of the matrix, including residual errors, which converged in an overidentified model that can clearly be identified, and hence a recursive model emerged; (2) for specification of the model, which was based on the factorial structure previously explained in the exploratory analysis, the latent variables were shown as ovals and the observed variables as rectangles, while the measurements and residual errors were shown as small ovals; (3) for estimation of the parameters, the AMOS program was used, with which the maximum likelihood method was applied using standardized estimators such as R^2^ (multiple squared correlations), the covariance of each estimator, the indices to be modified, and the critical proportions for the differences; (4) for evaluation of fit, all coefficients were rigorously reviewed, making sure that they were below acceptable limits (for example, that there were no negative or non-significant errors of variance, standardized coefficients greater than 1, or very high standard errors relative to some estimated coefficient; since the value of the correlations was less than 0.300, it could be verified that collinearity was not observed in the measured variables [35], or extreme or univariate scores [36]. Symmetry observed in all variables was excellent (values less than 1 ± 10 points) [37].

The models obtained from the confirmatory analysis are shown in Figure 1. For model 1, the absolute measurements of the total fit, the value of the chi-square (15.02, *p* = 306, 13 *df*), and the value of its adjusted complement (CMIN/DF = 1.155) were excellent, verifying the nullity in the errors of the variances and covariances of the model, which fits the sample, as recommended by experts [33]. 

It was observed that the comparative measures of global adjustment, both the Comparative Fit Index (CFI = 0.988), as well as the measure that considers the complexity of the model (Tucker–Lewis Index = 0.980), easily exceeded the ideal, indicating an excellent fit. The parsimony indices (PCFI = 0.612) and goodness of fit (GFI = 0.967, AGFI = 0.928) easily exceeded the acceptable limits, so the model was classified as complex [38]. The mean square root index of the approximation error (RMSEA = 0.036, CI95% = 0.000–0.101), which penalizes the increase in the complexity of the model, gave an error value well below the threshold of 0.08 and close to zero, confirming an almost perfect fit of the data model [33].

The confirmatory analysis was repeated with the remaining 50% of the randomly drawn data to compare the results in both subsamples. Table 2 shows this comparison, as well as the resulting indices for each calculated model.

### 3.2. Factors Associated with Low Appetite 

We used the second sample of 182 patients to evaluate the association of the final total appetite score (and the two factors of appetite) with other study variables (Table 3). The total appetite score and the two factors had significant negative correlations with the dialysis malnutrition score, depression, and anxiety scores, and a positive correlation with serum phosphorus levels. The total appetite score and factor 2 showed significant positive and negative correlations with serum albumin and distorted thoughts score, respectively. No significant correlations between appetite score and hemoglobin, calcium, or blood urea nitrogen (BUN) were found.

To identify those patients with reduced appetite, we used the ADAT appetite question as the gold standard, as described in the Methods section. In this sample of 182 patients, the responses to the ADAT question were: very good (N = 21, 11.5%), good (N = 105, 57.7%), fair (N = 42, 23.1%), poor (N = 8, 4.4%), and very poor (N = 6, 3.3%). We obtained a group of 56 patients with diminished appetite (which included those with responses of fair, poor, or very poor appetite in the ADAT question) and a group of 126 patients who had no reduced appetite (which included those with responses of good or very good appetite in the ADAT question). The ROC curve analysis of the total appetite score showed that the best cut-off value in the new appetite score for diminished appetite was ≤25 points, with a sensitivity of 73%, a specificity of 77%, an area under the curve (AUC) score of 0.86 (0.8 –0.9, CI 95%, *p* < 0.001), and an orthogonal distance of 0.35 to the optimum value (0,1) (Figure 2).

Table 4 shows the anthropometric variables, laboratory results, dialysis malnutrition scores, and psychological variables compared by appetite level. Patients with an appetite score ≤ 25 points (fair, poor, and very poor appetite) were considered to have a diminished appetite. This group of patients shows a higher proportion of females; lower educational level; less remunerated work; and lower creatinine, phosphorus, and blood urea nitrogen (BUN) compared to the group without disturbed appetite. Patients with diminished appetite also exhibited higher dialysis malnutrition, depression, and anxiety scores for both somatic and cognitive symptoms and distorted thoughts scores for all subscales (perfectionism, catastrophism, negative self-labeling, and dichotomous thinking). No differences regarding age, marital status, body mass index, percentage of ideal weight, mid-arm circumference, tricipital skinfold, mid-arm muscle circumference, normalized protein nitrogen appearance, the proportion of patients with PEW, the proportion of patients with diabetes mellitus, HD vintage, HD session time, albumin, hemoglobin, or calcium were observed between groups. 

The logistic regression analysis in Table 5 shows that after controlling for female sex, age, and diabetes mellitus, diminished appetite was independently associated with depression and anxiety symptoms and distorted thoughts score.

## 4. Discussion

The main contribution of this work was successfully adapting a Spanish language version of the Council on Nutrition Appetite Questionnaire to assess appetite in ESRD patients treated with hemodialysis, demonstrating the reliability and validity of the new questionnaire (Figure 3). This instrument preserves seven of the eight original items. 

Through the confirmatory analysis, we proved that the theoretical structural model fits satisfactorily the data gathered in our population according to the most important reliability and validity indices [33,35,36,37,38]. The data obtained from both random samples (for exploratory and confirmatory analysis) shows that factors 1 and 2 were strongly associated. We also concluded that the model is balanced and parsimonious, as was confirmed by the most robust indicators of the fit of the model’s structure (CFI, RMR, and RMSEA) [34]. Along with the creation of the questionnaire, this work showed that the majority of ESRD patients undergoing chronic HD with diminished appetite were females. We also found metabolic and psychological differences, such as lower creatinine, BUN, and phosphorus serum levels; and higher depression and anxiety (cognitive and somatic symptoms and distorted thinking scores, including catastrophism, dichotomous thinking, negative self-labeling, and perfectionism). Higher dialysis malnutrition scores was observed in patients belonging to the diminished appetite category as well.

The main motivation for developing a new tool to evaluate appetite loss was the urge to study patients’ perceptions associated with it. The ADAT question evaluates the appetite level with only one question [39,40], while an advantage of our new tool is that it allows us to evaluate two principal dimensions associated with appetite loss: factor 1 (related to the perceived flavor of the food) and factor 2 (related to the perception of gastric fullness). The study of these factors also permits us to explore the association of psychological appetite dimensions with maladaptive cognitive modifications, malnutrition, mood disorders (physical and cognitive symptoms), and metabolic alterations. We adapted the CNAQ, which was originally developed for elderly patients [21]. However, we managed to successfully apply it to HD patients, which are also a malnutrition-susceptible population. The distinctive features of this questionnaire will enable the inquiry of eating habits and psychological and metabolic factors associated with appetite, and perhaps isolate the phenomenon from cognitive distortion, as flavor-related factor 1 is not associated with cognitive distortion. Nevertheless, the total appetite scale and its two factors are associated with depression and anxiety symptoms, indicating a strong bond between depression and anxiety and appetite level. In fact, we excluded question 8 from the original CNAQ because its factorial load was included in both factor 1 and factor 2. Therefore, it was considered an imprecise question to assess either factor.

Appetite assessment in ESRD patients is a complex task due to several potential confounders inherent to the disease and its complications. Appetite modification is one of the symptoms related to depression, anxiety, and uremia. Additionally, the interactions among these symptoms synergically modify patients’ motivation to eat. The best method to assess symptoms of depression in this population has been the object of debate. The BDI is a well-grounded method to achieve this purpose in HD patients, using a higher cut-off score from the general population [41,42,43,44]. Anxiety symptoms are less studied compared to depression, although they are highly prevalent (approximately 22% measured by BAI score) compared with the estimated 7.3% prevalence of anxiety disorders in the general population [45], and are related to higher mortality in these patients [46]. It is important to point out that even if the symptoms of mood and anxiety disorders also overlap, they belong to different categories of mental illness and require focused management. Interestingly, we found a larger proportion of females in the group with disturbed appetite in our study, which had higher depressive and anxiety symptoms scores. It is well known that depression and anxiety disorders are more prevalent in women than men [47,48]. Despite the identification of biological and physiological factors related to a higher prevalence of depression and anxiety, the reason for this difference remains unknown [49,50]. Moreover, advanced age is a well-recognized risk factor for developing depression [51], although we found no differences between groups in our study regarding age. 

Depression modifies appetite levels via immune system dysregulation and endocrine and nervous system imbalance [52], which leads to different subgroups of appetite modifications [53]. In this work, we focused on decreased appetite levels, as they pose a threat to the health of patients undergoing HD. Appetite reduction is related to increased death risk [4], hospitalization rates, diminished quality of life, elevated serum concentrations of pro-inflammatory cytokines, and erythropoietin hyperresponsiveness [4]. Depressive symptoms can be divided into cognitive and somatic symptoms, which refer to different psychological dimensions and are of great importance in clinical practice. In fact, affective (cognitive) symptoms of depression predict long-term mortality in patients treated with hemodialysis [16]. We found higher scores for both somatic and cognitive symptoms in patients with diminished appetite for depression and anxiety. As was demonstrated by multivariate analysis controlled by sex and age in this work, appetite loss was strongly associated with total depression and anxiety score. 

Distorted thinking is associated with depression and anxiety symptoms in patients undergoing HD and other chronic diseases [18,54,55]. In a previous study [19], we explored the prevalence of distorted thoughts in our population and developed a psychological intervention to improve this and other psychological aspects of patients undergoing HD. The present study shows that cognitive distortion is strongly associated with diminished appetite in univariate analysis. Additionally, this pattern was observed after multivariate analysis, suggesting that cognitive distortion modifies appetite on its own, as is observed in other psychopathologies [56], such as anorexia nervosa [57], mood disorders [58], and chronic diseases [59], but is often a neglected phenomenon in ESRD.

Patients undergoing HD are subjected to several lifestyle changes, such as dietary and fluid restrictions due to their medical condition. Their adherence to the treatment is highly influenced by the patients’ environment, perceptions, and social support [60,61,62,63]. The explicit role of distorted thinking in dietary and fluid restriction adherence is unknown. Interestingly, cognitive intervention therapy, which aims to cognitive restructuring and relieve maladaptive thoughts [64], has a positive effect on adherence to HD fluid restrictions [65]. 

Elevated serum phosphorus is a well-known risk factor for cardiovascular and all-cause mortality in patients with ESRD [66,67,68]. We also found lower levels of serum phosphorus in patients with diminished appetite. However, we did not observe a significant correlation with diminished appetite after multivariate analysis. There was no difference in albumin levels between patients with normal and diminished appetite, despite its correlations with total appetite and factor 2 score. This is probably attributed to the relatively moderate severity of malnutrition in our population. Creatinine levels were lower in HD patients with diminished appetite within our study, as were BUN levels. Creatinine levels are influenced by nutrition, inflammation, serum protein, and muscle mass [4,69]. It is likely that creatinine and BUN are decreased by malnutrition and muscle mass loss. However, neither creatinine nor BUN were associated with the diminished appetite score after multivariate analysis. Additional limitations of this work include the potential influence of cardiovascular, cerebrovascular, and peripheral vascular disease and gastrointestinal disturbances, which were not assessed. Furthermore, we did not explore other potential associated factors, such as total iron-binding capacity, subcutaneous fat, and muscle wasting. 

## 5. Conclusions

The adapted appetite score adequately discriminates against patients with diminished appetite and discerns two main components of appetite in HD patients—perceived flavor of food and perception of gastric fullness. Low scores on this scale are associated with elevated dialysis malnutrition scores. Moreover, depression, anxiety, and distorted thinking are independent factors of diminished appetite. Screening and treatment for psychological conditions may be useful to increase appetite and the nutritional status of these patients. 

## Figures and Tables

**Figure 1 nutrients-13-01371-f001:**
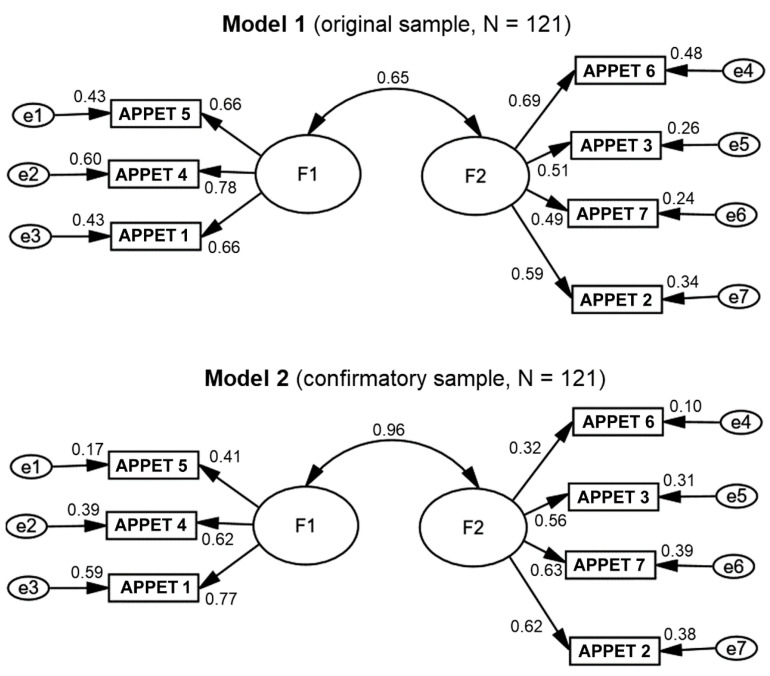
Confirmatory analysis models. F1 = factor 1 (related to the perceived flavor of the food); F2 = factor 2 (related to the perception of gastric fullness). Items: APPET 1 = “My appetite is…”; APPET 2 = “When I eat…”; APPET 3 = “I feel hungry…”; APPET 4 = “Food tastes…”; APPET 5 = “Compared to when I was younger, food tastes…”; APPET 6 = “Normally I eat…”; APPET 7 = “I feel sick or nauseated when I eat…”;. APPET refers to each one of the items of the appetite questionnaire.

**Figure 2 nutrients-13-01371-f002:**
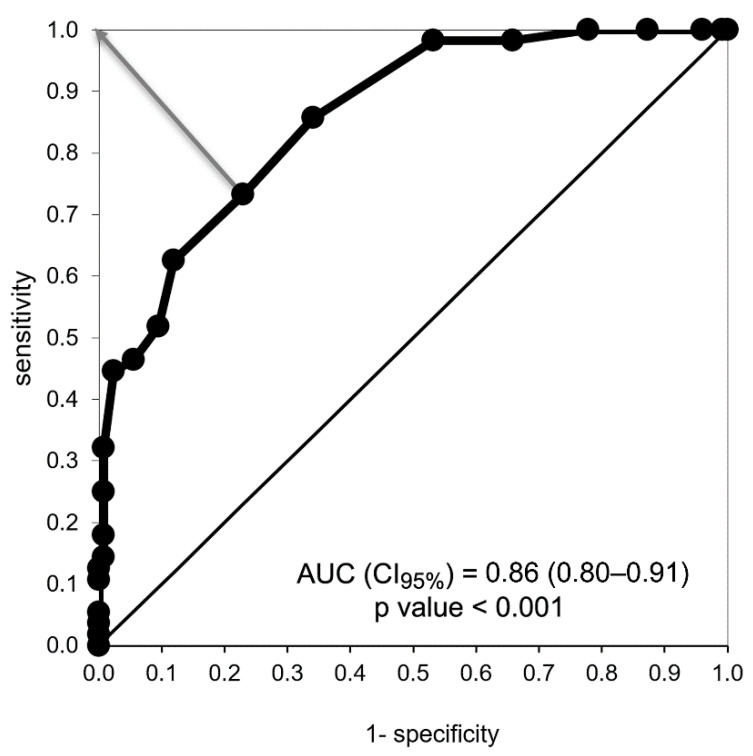
Receiver operator characteristic (ROC) curve analysis of the total appetite score to determine the optimum cut-off value for low appetite. An answer of “poor”, “very poor”, or “regular” appetite in the ADAT question was considered as the reference value. The best value to identify those with low appetite was a total appetite score ≤ 25 points, with a sensitivity of 73%, specificity of 77%, and an orthogonal distance of 0.35 to the optimum value (0,1). The area under the curve (AUC) score was 0.86 (0.8–0.9, CI 95%, *p* < 0.001)

**Figure 3 nutrients-13-01371-f003:**
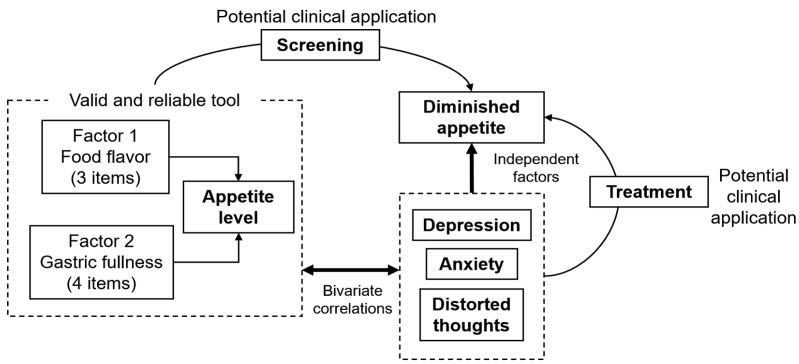
Diagram of the main contributions of the current study. The adapted questionnaire is a simple, valid, and reliable tool to assess appetite level in Spanish-speaking patients treated with hemodialysis. The ROC curve analysis showed that a low score discriminates against those patients with diminished appetite, with a potential application as a screening tool. Depression symptoms, anxiety symptoms, and distorted thoughts are independent factors of diminished appetite. Treatment of these psychological variables may be useful to improve appetite level.

**Table 1 nutrients-13-01371-t001:** Exploratory factor analysis of The Council on Nutrition Appetite Questionnaire (CNAQ) in a random sample of 121 Mexican patients with end-stage renal disease on hemodialysis. Total Cronbach’s alpha = 0.758; total mean = 24.6 ± 4; total variance = 16.1; Hotelling’s *t*-squared test, F = 376.6, 116 *df*, *p* ≤ 0.000; Total explained variance = 57.6%.

Item	Factorial Load		Item Mean ± SD
	Factor 1 (3 Items)	Factor 2 (4 Items)	
APPET 5 “Compared to when I was younger, food tastes…”	0.885		2.83 ± 0.68
APPET 4 “Food tastes…”	0.783		3.79 ± 0.84
APPET 1 “My appetite is…”	0.667		3.59 ± 0.97
APPET 6 “Normally I eat…”		0.737	3.66 ± 0.83
APPET 3 “I feel hungry…”		0.704	2.96 ± 1.06
APPET 7 “I feel sick or nauseated when I eat…”		0.681	4.20 ± 1.04
APPET 2 “When I eat…”		0.558	3.59 ± 0.87
Alpha value of the factor	0.739	0.656	
Percentage of explained variance	28.9	28.7	
Mean	10.20	14.40	
Standard deviation	2.01	2.66	
Factor variance	4.05	7.09	
Intraclass factor correlation	0.724	0.644	
Lower value	0.627	0.529	
Higher value	0.799	0.737	
F value	3.63	2.81	
*p* value	≤0.001	≤0.001	

APPET refers to each one of the items of the appetite questionnaire; SD = standard deviation.

**Table 2 nutrients-13-01371-t002:** Goodness of fit indices of the confirmatory model resulting from 2 factors in kidney patients (*n* = 242) using the method of maximum likelihood.

Statistics	Desirable Criterion	Model #1 (*n* = 122)	Model #2 (*n* = 122)	Interpretation
Absolute fit X2/*df* (CMIN/*df*)	Less than 2 or 3	(CMIN/*df*) = 1.155	(CMIN/*df*) = 1.239	The errors of the model are null with the sample used and the absolute fit is excellent
Goodness of fit index (GFI)	>0.900 Preferential > 0.950	GFI = 0.967	GFI = 0.964	Good fit
Comparative goodness of fit index (CFI)	>0.900, Preferential > 0.950	CFI = 0.988	CFI = 0.980	Acceptable comparative fit
Root mean square residual (RMR)	Near zero	RMR = 0.040	RMR = 0.036	Model error close to zero, almost perfect fit of model to data
Root mean square error of approximation (RMSEA)	Less than 0.08, close to zero	RMSEA = 0.036 (0.000–0.101)	RMSEA = 0.045 (0.000–0.106)	Model error close to zero, almost perfect fit of model to data

X2 = Chi-squared test; *df* = degrees of freedom; CMIN = Chi-squared model index.

**Table 3 nutrients-13-01371-t003:** Spearman’s correlation analysis between the appetite score, malnutrition variables, and psychological variables (*n* = 182).

	Appetite Score		
Variables	Total Score	Factor 1	Factor 2
Dialysis malnutrition score	−0.227 **	−0.217 **	0.154 *
Albumin (g/dL)	0.258 **	0.135	0.294 **
Creatinine (mg/dL)	0.116	0.121	0.082
Hemoglobin (mg/dL)	0.039	0.012	0.074
Phosphorus (mg/dL)	0.252 **	0.173 *	0.217 **
Calcium (mg/dL)	−0.040	−0.066	−0.008
BUN (mg/dL)	0.108	0.034	0.122
Total depression score	−0.372 **	−0.313 **	−0.339 **
Total anxiety score	−0.362 **	−0.310 **	−0.317 **
Distorted thoughts score	−0.222 **	−0.133	−0.238 **

Note: * *p* < 0.05; ** *p* < 0.01.

**Table 4 nutrients-13-01371-t004:** Anthropometric variables, laboratory results, dialysis malnutrition scores, and psychological variables compared by appetite level in 182 ESRD patients treated with chronic HD.

	Diminished Appetite Score (≤25 points)		
	Yes (N = 56)	No (N = 126)	*p*-Value
Age (years)	52 (40–62)	48 (34–58)	0.064
Female sex	31 (55%)	42 (33%)	0.008
*Educational level*			0.021
Elementary	48 (86%)	89 (71%)	
High school or higher	8 (14%)	37 (29%)	
*Marital status*			0.837
Single	18 (32%)	39 (31%)	
Couple	38 (68%)	87 (69%)	
Remunerated work	8 (14%)	37 (29%)	0.021
Diabetes mellitus	25 (45%)	54 (43%)	0.872
HD vintage (months)	24 (9–42)	24 (9–48)	0.780
HD session time (hours)	3.8 (3.0–4.0)	3.5 (3.0–4.0)	0.983
Albumin (g/dL)	3.7 (3.4–4.1)	3.8 (3.4–4.3)	0.496
Creatinine (mg/dL)	8.6 (6.8–10.4)	9.9 (7.5–12)	0.025
Hemoglobin (mg/dL)	9.4 (7.9–10.4)	8.9 (7.7–10.4)	0.443
Phosphorus (mg/dL)	4.6 (3.8–6.7)	5.8 (4.5–7.5)	0.019
Calcium (mg/dL)	8.9 (8.3–9.4)	8.7 (8.1–9.4)	0.554
BUN (mg/dL)	54 (47–69)	64 (50–76)	0.038
Body mass index (Kg/m^2^)	23.3 (21.6–27.9)	24.2 (21.4–29.1)	0.497
Percentage of ideal weight	101.6 (93.9–114.3)	105.6 (93.3–121.4)	0.408
Mid-arm circumference (cm)	25.5 (23.3–28.4)	26.5 (24.0–29.5)	0.252
Tricipital skinfold (cm)	1.3 (0.9–1.6)	1.2 (0.9–1.6)	0.977
Mid-arm muscle circumference	21.6 (19.5–23.7)	22.8 (20.7–24.7)	0.074
nPNA (g/kg/day)	1.12 (0.83–1.35)	1.16 (0.87–1.48)	0.412
Dialysis malnutrition score	16 (13–19)	14 (12–17)	0.032
PEW (%)	15 (27%)	27 (21%)	0.429
*Total depression score*	15 (7–23)	7 (3–12)	<0.001
Somatic symptoms	11 (6–16)	6 (2–9)	<0.001
Cognitive symptoms	3 (1–8)	1 (0–4)	0.001
*Total anxiety score*	15 (7–23)	7 (3–12)	<0.001
Somatic symptoms	8 (3–15)	4 (1–7)	<0.001
Cognitive symptoms	3 (1–5)	0 (0–3)	<0.001
*Distorted thoughts score*	56 (48–72)	44 (37–59)	<0.001
Catastrophism	23 (16–28)	15 (12–25)	0.001
Dichotomous thinking	16 (10–20)	11 (9–25)	0.046
Negative self-labelling	9 (7–13)	7 (6–11)	0.001
Perfectionism	8 (6–11)	5 (5–7)	<0.001

Note: ESRD = end-stage renal disease; HD = hemodialysis; BUN = blood urea nitrogen; nPNA = normalized protein nitrogen appearance; PEW = protein energy waste syndrome.

**Table 5 nutrients-13-01371-t005:** Logistic regression analysis of factors related to diminished appetite (total score ≤ 25 points) in 182 ESRD patients treated with HD.

	Univariate		Multivariate	
	O.R. (I.C.95%)	*p*	O.R. (I.C.95%)	*p*
Age (years)	1.02 (1.00–1.04)	0.08	1.02 (1.00–1.05) *	0.05
Female sex	2.48 (1.30–4.72)	<0.01	2.66 (1.36–5.17) *	<0.01
Elementary education	2.49 (1.08–5.78)	0.03	1.97 (0.77–5.00) ^&^	0.15
Remunerated work	0.40 (0.17–0.93)	0.03	0.50 (0.20–1.28) ^&^	0.15
Dialysis malnutrition score	1.07 (0.99–1.17)	0.10	1.04 (0.95–1.13) ^&^	0.43
Creatinine (mg/dL)	0.92 (0.85–1.00)	0.06	0.96 (0.88–1.06) ^&^	0.42
Phosphorus (mg/dL)	0.85 (0.72–0.99)	0.05	0.86 (0.73–1.01) ^&^	0.07
BUN (mg/dL)	0.99 (0.98–1.00)	0.09	0.99 (0.97–1.00) ^&^	0.10
*Total depression score*	1.09 (1.05–1.13)	<0.01	1.08 (1.04–1.13) ^&^	<0.01
Somatic symptoms	1.15 (1.07–1.22)	<0.01	1.14 (1.06–1.22) ^&^	<0.01
Cognitive symptoms	1.11 (1.08–1.23)	<0.01	1.16 (1.06–1.27) ^&^	<0.01
*Total anxiety score*	1.11 (1.05–1.16)	<0.01	1.10 (1.04–1.15) ^&^	<0.01
Somatic symptoms	1.14 (1.07–1.22)	<0.01	1.13 (1.06–1.21) ^&^	<0.01
Cognitive symptoms	1.23 (1.09–1.38)	<0.01	1.20 (1.06–1.35) ^&^	<0.01
*Distorted thoughts score*	1.03 (1.02–1.05)	<0.01	1.03 (1.02–1.06) ^&^	<0.01

* The odds ration (O.R.) values were obtained from one model, which included age and female sex. ^&^ The O.R. values were obtained from a model adjusted by age and female sex. BUN = blood urea nitrogen

## Data Availability

The data presented in this study are available on request from the corresponding author. The data are not publicly available to protect the study participants from potential identification based on their personal information provided in the study variables.

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
