# Peer review of "Assessment of Factors Related to Diminished Appetite in Hemodialysis Patients with a New Adapted and Validated Questionnaire"

_nutrients, 2021, doi:10.3390/nu13041371_

Round 1
Reviewer 1 Report
The article entitled “Assessment of factors related to diminished appetite in hemodialysis patients with a new adapted and validated questionnaire” is a current and interesting topic in hemodialysis patients. Some comments of the manuscript are suggested to the authors.
In order to apply this validated questionnaire, the final version should be included as a supplementary material.
Line 81. It is important to modify this issue “For malnutrition assessment” since DMS is a nutritional screening score. It is different concept.
Lines 97-12. Theses lines should be transferred to the statistical analysis section
In material and method section, there is not any comment about anthropometry or lab parameters used in this study. It must be explained in this section
Results. Which is the prevalence of PEW in this sample?. It should be defined. Furthermore, the overall socio-demographic characteristics of the sample have not been included in Table 4. It is mandatory.
Figure 2. Image quality needs to be improved
Author Response
Manuscript ID: nutrients-1172477 “Assessment of factors related to diminished appetite in hemodialysis patients with a new adapted and validated questionnaire”
Response to comments from Reviewer 1
Comment: The article entitled “Assessment of factors related to diminished appetite in hemodialysis patients with a new adapted and validated questionnaire” is a current and interesting topic in hemodialysis patients. Some comments of the manuscript are suggested to the authors.In order to apply this validated questionnaire, the final version should be included as a supplementary material.
Response: We appreciate the positive reviewer’s commentsand helpful suggestions. All changes in the manuscript are highlighted with red color. The final version of the validated questionnaire is now included in the Appendix A.
Comment: Line81. It is important to modify this issue “For malnutrition assessment” since DMS is a nutritional screening score. It is different concept.
Response: It is, indeed, a different concept. This semantic error has been corrected in the new version of the manuscript (line 90).
Comment: Lines 97-12. Theses lines should be transferred to the statistical analysis section
Response: The suggested lines were transferred to the statistical analysis section (lines 136to 152).
Comment: In material and method section, there is not any comment about anthropometry or lab parameters used in this study. It must be explained in this section.
Response: Details of the anthropometry and laboratory parameters are now mentioned in the Material and Method section (lines 79to 86).
Comment: Results. Which is the prevalence of PEW in this sample? It should be defined. Furthermore, the overall socio-demographic characteristics of the sample have not been included in Table 4. It is mandatory.
Response: The prevalence of PEW in this sample was 23.1%, with no significant differences between groups according to the diminished appetite score (Table 4).We have included the overall socio-demographic characteristics of the participants.
Comment: Figure 2. Image quality needs to be improved
Response: The image quality of Figure 2 has been improved.
Reviewer 2 Report
The manuscript describes an intersting study aiming to adapt and validate a Spanish version of the Council on Nutrition Appetite Questionnaire (CNAQ) to a population of HD patients. THe study shows that this tool was able to discriminate patients with diminished appetite and that low scores corelated with dialysis malnutrition parameters.
The study is well conducted and described. However, major finidings confirm data already known and described in the literature. The authors should underlie the novelty of the study in the discussion section/show a figure synthetizing what's new with your sudy..something that may increase the interst of the readers.
Author Response
Manuscript ID: nutrients-1172477 “Assessment of factors related to diminished appetite in hemodialysis patients with a new adapted and validated questionnaire”
Response to comments from Reviewer 2
Comment: The manuscript describes an intersting study aiming to adapt and validate a Spanish version of the Council on Nutrition Appetite Questionnaire (CNAQ) to a population of HD patients. THe study shows that this tool was able todiscriminate patients with diminished appetite and that low scores corelated with dialysis malnutrition parameters.The study is well conducted and described. However, major finidings confirm data already known and described in the literature. The authors should underlie the novelty of the study in the discussion section/show a figure synthetizingwhat's new with your sudy..something that may increase the interst of the readers.
Response: We thank the reviewer for these positive comments about our manuscript. We have added a diagram in the Discussion section to emphasize the maincontributions of our work (Figure 3). Moreover, a Graphical Abstract is now included in the revised submission of our manuscript.